

# Expression and purification of human diacylglycerol kinase α from baculovirus-infected insect cells for structural studies

Daisuke Takahashi and Fumio Sakane

Department of Chemistry, Graduate School of Science, Chiba University, Chiba, Japan

## ABSTRACT

Diacylglycerol kinases (DGKs) are lipid kinases that modulate the levels of lipid second messengers, diacylglycerol and phosphatidic acid. Recently, increasing attention has been paid to its α isozyme (DGKα) as a potential target for cancer immunotherapy. DGKα consists of the N-terminal regulatory domains including EF-hand motifs and C1 domains, and the C-terminal catalytic domain (DGKα-CD). To date, however, no structures of mammalian DGKs including their CDs have yet been reported, impeding our understanding on the catalytic mechanism of DGKs and the rational structure-based drug design. Here we attempted to produce DGKα-CD or a full-length DGKα using bacterial and baculovirus-insect cell expression system for structural studies. While several DGKα-CD constructs produced using both bacterial and insect cells formed insoluble or soluble aggregates, the full-length DGKα expressed in insect cells remained soluble and was purified to near homogeneity as a monomer with yields (1.3 mg/mL per one L cell culture) feasible for protein crystallization. Following enzymatic characterization showed that the purified DGKα is in fully functional state. We further demonstrated that the purified enzyme could be concentrated without any significant aggregation, and characterized its secondary structure by circular dichroism. Taken together, these results suggest that the presence of N-terminal regulatory domains suppress protein aggregation likely via their intramolecular interactions with DGKα-CD, and demonstrate that the baculovirus-insect cell expression of the full-length form of DGKα, not DGKα-CD alone, represents a promising approach to produce protein sample for structural studies of DGKα. Thus, our study will encourage future efforts to determine the crystal structure of DGK, which has not been determined since it was first identified in 1959.

Corresponding author
Fumio Sakane,
sakane@faculty.chiba-u.jp

## INTRODUCTION

Diacylglycerol (DG) and phosphatidic acid (PA) are important signaling lipids and regulate a myriad of cellular events by modulating numerous signaling proteins (*English, 1996*; *Stace & Ktistakis, 2006*; *Griner & Kazanietz, 2007*; *Almena & Merida, 2011*),

including protein kinase C isoforms (*Newton, 1997*; *Parekh, Ziegler & Parker, 2000*; *Griner & Kazanietz, 2007*) and Ras guanyl nucleotide-releasing protein (RasGRP) (*Ebinu et al., 1998*; *Tognon et al., 1998*) by DG, and mammalian target of Rapamycin (*Ávila-Flores et al., 2005*) and phosphatidylinositol (PI)-4-phosphate 5-kinase (*Moritz et al., 1992*) by PA. Diacylglycerol kinases (DGKs), which was first identified in 1959 (*Hokin & Hokin, 1959*), are a family of lipid kinase that regulates the intracellular levels of DG and PA by phosphorylating DG into PA (*Sakane et al., 2007*; *Merida, Ávila-Flores & Merino, 2008*; *Shulga, Topham & Epand, 2011*). Mammalian DGK consists of 10 isozymes ($\alpha$, $\beta$, $\gamma$, $\delta$, $\eta$, $\kappa$, $\varepsilon$, $\zeta$, $\iota$, and $\theta$), classified in five subtypes featuring distinct regulatory domains and a conserved catalytic domain (CD; *Sakane et al., 2007*; *Shulga, Topham & Epand, 2011*), and each DGK serves as a key downregulator and upregulator for the DG and PA-mediated cellular signaling, respectively.

Diacylglycerol kinase $\alpha$ is the first-cloned DGK isozyme in mammals (*Sakane et al., 1990*) and has amino-terminal regulatory domains including EF-hand motifs and C1 domains, and a carboxyl-terminal CD (Fig. 1A). Recently, increasing attention has been paid to DGK$\alpha$ as a potential target for anti-cancer treatments including cancer immunotherapy (*Dominguez et al., 2013*; *Purow, 2015*; *Sakane, Mizuno & Komenoi, 2016*; *Liu et al., 2016*; *Noessner, 2017*). Expression of DGK$\alpha$ has been reported to be upregulated in melanoma cells (but not in noncancerous melanocytes) (*Yanagisawa et al., 2007*), lymphoma (*Bacchiocchi et al., 2005*), hepatocellular carcinoma (*Takeishi et al., 2012*), breast cancer cells (*Torres-Ayuso et al., 2014*), and glioblastoma cells (*Dominguez et al., 2013*) where DGK$\alpha$ promotes cancer cell survival, proliferation, migration, and invasion (*Merida et al., 2017*). siRNA knockdown of *DGKA* or inhibition of DGK$\alpha$ by small molecule inhibitors for DGKs, R59022 and R59949, has detrimental effects on the proliferation of glioblastoma cells, melanoma, breast cancer, and cervical cancer cells (*Yanagisawa et al., 2007*; *Dominguez et al., 2013*). In T-lymphocytes, on the other hand, DGK$\alpha$ is appreciated as a critical attenuator for immune response. DGK$\alpha$ is highly expressed in T-cells and decreases membrane DG levels required for RasGRP1-dependent activation of the Ras–Erk pathway (*Jones et al., 2002*). Furthermore, in vitro and in vivo studies have uncovered that DGK$\alpha$ is responsible for T-cell hyporesponsive state known as anergy state (*Olenchock et al., 2006*; *Zha et al., 2006*).

Using a high-throughput DGK assay, we have recently identified a novel DGK$\alpha$-selective inhibitor, CU-3, and revealed that this compound targets the CD of DGK$\alpha$ (*Liu et al., 2016*). Indeed, this compound not only induced the apoptosis of HepG2 hepatocellular carcinoma and HeLa cervical cancer cells as observed for other DGK inhibitors with lower-selectivity (*Dominguez et al., 2013*), but also enhanced the production of interleukin-2 in Jurkat T cells (*Liu et al., 2016*), illustrating a double-strike effect of DGK$\alpha$ inhibitors potentially utilized for cancer immunotherapy (*Noessner, 2017*). However, despite these biological and biomedical importance, no structure has been determined for the CDs of any mammalian DGKs, thus impeding the detailed understanding of DGK catalytic machinery and substrate binding sites as well as the development and optimization of effective DGK$\alpha$ inhibitors.

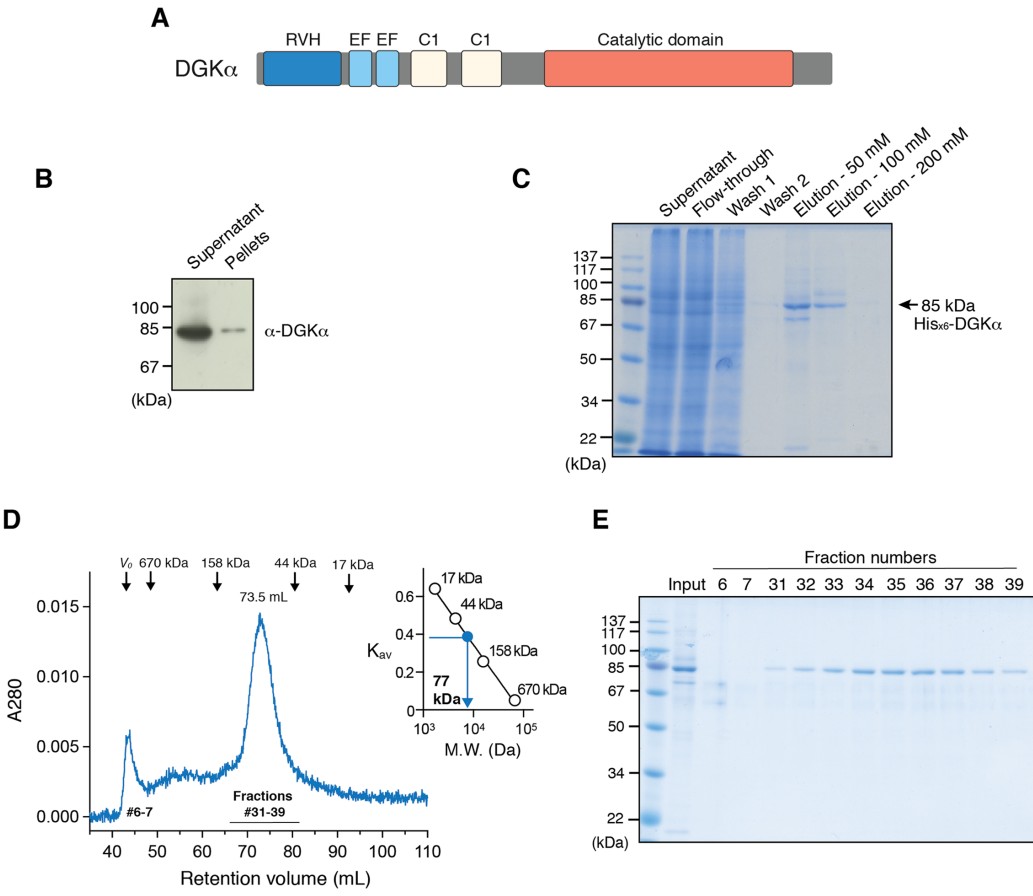

**Figure 1 Expression of DGKα in baculovirus-infected insect cells and purification.** (A) Domain architecture of DGKα. RVH, recoverin homology domain; EF, EF-hand motif; C1, cysteine-rich C1 domain. (B) Immunoblot analysis of the solubility of a full-length DGKα expressed in Sf9 cells. Cell lysates were separated into supernatant and insoluble pellets and subjected to SDS-PAGE (10%) followed by immunoblot analysis using anti-DGKα antibody. (C) SDS-PAGE (10%) analysis of fractions from $Ni^{2+}$-affinity purification. Separated proteins were stained with Coomassie blue staining. (D) Elution profile of DGKα from size exclusion chromatography. Fraction numbers used for following SDS-PAGE analysis are labeled. The inset shows the calibration of gel-filtration column using protein standards of known molecular weight (thyroglobulin (670 kDa), γ-globulin (158 kDa), ovalbumin (44 kDa), myoglobin (17 kDa)). Partition coefficient ($K_{av}$) was calculated from the formula, $K_{av} = (V_E - V_0)/(V_T - V_0)$, where $V_E$ is the retention volume of each sample, $V_T$ is the total column volume (120 mL), and $V_0$ is the void volume of the column (44 mL), respectively. $K_{av}$ was plotted against the molecular weight of proteins and linear regression analysis was conducted. (E) SDS-PAGE (10%) analysis of DGKα purified using size-exclusion chromatography.

One of the greatest challenges for the structure determination of DGK isozymes lies in producing enough and soluble proteins suitable for protein crystallization, as illustrated in previous studies (*Takahashi et al., 2012*; *Petro & Raben, 2013*). Although intensive efforts have been made by *Petro & Raben (2013)* to express and purify the full-length and CD of porcine DGKα using bacterial expression system with expression-tags (glutathione S-transferase (GST), maltose binding protein (MBP), thioredoxin (TRX)) for solubility enhancement, all the expressed DGKα constructs formed inclusion bodies or soluble aggregates, likely due to the inability of bacterial translational and folding machineries.

To overcome these problems, here we have taken advantage of the baculovirus-insect cell expression system to express a full-length DGKα in soluble form. DGKα expressed in the insect cells was then purified by a series of column chromatography, and the purified protein was found to be a monomer in solution. Successful purification of DGKα also allowed us to characterize enzymatic, inhibitory and structural properties of DGKα in vitro. Taken together, these results provide promising evidence that the baculovirus-insect cell expression system is better suited to produce DGKα for in vitro functional and structural studies.

## MATERIALS AND METHODS

### Bacterial expression and purification of DGKα-CD

Multiple-constructs approach with different N- and C-terminal boundaries (*Gräslund et al., 2008*), and several N-terminal fusion-tags (GST, MBP, and small ubiquitin-like modifier (SUMO)) was applied for bacterial expression of DGKα-CD.

To prepare GST-fused constructs, the DNA sequences of DGKα-CD (S332–G722, D344–G722, D369–G722, S332–S735, D344–S735, D369–S735), flanked by *Bam*HI and *Sal*I restriction sites were amplified by PCR from the full-length cDNA for human DGKα, inserted into a pGEX-4T-2 vector (GE Healthcare Life Science, Little Chalfont, UK) and the resulting plasmids were used to transform *Escherichia coli* strain Rosetta2 (DE3) (Novagen, Madison, WI, USA). The protein construct contained a thrombin-cleavable GST-tag before the DGKα-CD sequence. Cells were cultured in LB media at 37 °C until $OD_{600}$ reached 0.6–0.8. Expression of the recombinant protein was then induced by adding 0.5 mM isopropyl β-D-thiogalactopyranoside (IPTG), and the bacterial culture was continued at 16 °C for overnight. Bacteria harvested by centrifugation were suspended in a lysis buffer (50 mM sodium phosphate, pH 8.0, containing 500 mM NaCl, 1 mM phenylmethylsulfonyl fluoride, 1 mM dithiothreitol) and lysed by sonication on ice. Protease inhibitors (20 μg/mL aprotinin, 20 μg/mL leupeptin, 20 μg/mL pepstatin, 1 mM soybean trypsin inhibitor) were added immediately before sonication. To evaluate expression and solubility of the expressed proteins, soluble and insoluble fractions were separated by centrifugation at 15,000×g for 10 min and subjected to SDS-PAGE (10%) followed by Coomassie Brilliant Blue (CBB) staining and immunoblot analysis using anti-GST monoclonal antibody (B-14; Santa Cruz Biotechnology, Dallas, TX, USA). The immunoreactive bands were visualized using peroxidase-conjugated anti-mouse IgG antibodies (Jackson ImmunoResearch Laboratories, West Grove, PA, USA) and the ECL Western blotting detection system (GE Healthcare Life Science, Little Chalfont, UK).

For Sumo-tag, the DNA sequences of DGKα-CD (S332–G722, D344–G722, D369–G722, S332–S735, D344–S735, D369–S735) were cloned via *Nde*I/*Sal*I sites into pSUMO vector. In addition, three constructs with additional two glutamic acids at the C-terminus (S332–S735EE, D344–S735EE, D369–S735EE) were also cloned into the pSUMO vector. The resulting recombinant protein contains an N-terminal His-tagged Sumo domain, followed by a Sumo-specific protease (Ulp1) cleavage site before the DGKα-CD sequence. $His_{×6}$-Sumo-DGKα-CD was expressed in *E. coli* Rosetta2 (DE3) cells by induction with 0.1 mM IPTG at 16 °C for overnight. After a small-scale expression

and solubility test using anti-His$_{\times 6}$ monoclonal antibody (9C11; Wako, Osaka, Japan), large scale expression of the construct D344–S735EE was conducted. After cell-lysis and centrifugation, Ni-affinity chromatography was applied to purify His$_{\times 6}$-Sumo-DGKα-CD (D344–S735EE). The column was washed with 50 mM Tris–HCl, pH 8.0, 500 mM NaCl, 10 and 50 mM imidazole, and the bound proteins were eluted with 300 mM imidazole. An elution fraction containing His$_{\times 6}$-Sumo-DGKα-CD was concentrated using a centrifugal filter (Amicon Ultra-15; Millipore, Burlington, MA, USA), and applied to a Superdex 75 16/60 column for size exclusion chromatography purification.

Maltose binding protein-fused DGKα-CD constructs were prepared by cloning the sequences of DGKα-CD (S332–G722, D344–G722, D369–G722, S332–S735, D344–S735, D369–S735, S332–S735EE, D344–S735EE, D369–S735EE) into a pMAL-c2X vector (New England Biolabs, Ipswich, MA, USA) via *Bam*HI/*Sal*I sites. *E. coli* strain Rosetta2 (DE3) transformed with the plasmids. Protein expression was induced with 0.1 mM IPTG and bacterial cells were then incubated at 16 °C for overnight. After cell-lysis and centrifugation, the expression and solubility test was conducted by subjecting soluble and insoluble fractions on SDS-PAGE (10%) followed by CBB staining and immunoblot analysis using anti-MBP antibody (Sc-13564; Santa Cruz Biotechnology, Dallas, TX, USA). The construct MBP-DGKα-CD (D369–S735) was expressed in 2 L of LB medium and the MBP-fused protein was purified by affinity chromatography on amylose resin (New England Biolabs, Ipswich, MA, USA). The affinity column was washed with a buffer, 20 mM Tris–HCl, pH 7.4, 200 mM NaCl, 1 mM phenylmethylsulfonyl fluoride, 1 mM EDTA, and the bound proteins were eluted with a buffer containing 10 mM maltose. Fractions were subjected to SDS-PAGE (10%) and analyzed by CBB staining and immunoblotting using anti-MBP antibody and anti-DGKα antibody (*Yamada, Sakane & Kanoh, 1989*).

### Expression of DGKα in insect cells using baculovirus expression vector system

The construct of DGKα-CD (D364–S735) or full-length DGKα with N-terminal His$_{\times 6}$ tag was PCR-amplified and cloned into the pOET3 vector (Oxford Expression Technologies, Oxford, UK) via *Sal*I/*Not*I sites. The resulting DNA sequences were verified to be correct by DNA sequencing. The flashBAC system (Oxford Expression Technologies, Oxford, UK) was used to generate a recombinant baculovirus and the virus stock was amplified by several rounds of infection of Sf9 cells cultured in Sf-900 II serum free medium (Invitrogen, Carlsbad, CA, USA) at a low multiplicity of infection (MOI). Plaque assays were performed to determine titers of the amplified virus stocks. Both DGKα-CD and full-length DGKα were expressed in Sf9 cells by infecting the cells (at $2 \times 10^6$ cells/mL) with the baculovirus stock at MOI of 2. Cells were cultured at 28 °C with shaking for 24 h and pelleted by centrifugation at 1,500×g, 4 °C for 20 min and washed with sterile phosphate buffered saline before storage at −80 °C.

### Purification of DGKα expressed in insect cells

The cell pellets were thawed and suspended in a lysis buffer containing 50 mM Tris–HCl, pH 8.0, 0.5M NaCl, 20 mM imidazole, 20% glycerol, 5 mM CaCl$_2$, 5 mM MgCl$_2$,

5 mM β-mercaptoethanol, 1% Nonidet P-40 (NP-40), 5 mM adenosine 5'-diphosphate (ADP), 5 U/mL benzonase (EMD Millipore, Burlington, MA, USA) and a EDTA-free protease inhibitor cocktail tablet (Roche, Basel, Switzerland). The cell suspension was lysed by sonication on ice followed by centrifugation at $25,000 \times g$, 4 °C for 1 h. The supernatant was incubated with 2 mL of Ni-NTA agarose (Qiagen, Venlo, The Netherlands) for 2 h at 4 °C, and then the mixture was packed into a column by gravity. The column was washed with wash buffer 1 (50 mM Tris–HCl, pH 8.0, 0.5M NaCl, 20 mM imidazole, 20% glycerol, 5 mM $CaCl_2$, 5 mM $MgCl_2$, 5 mM β-mercaptoethanol, 1% NP-40) and wash buffer 2 (50 mM Tris–HCl, pH 8.0, 0.5M NaCl, 20 mM imidazole, 20% glycerol, 5 mM $CaCl_2$, 5 mM $MgCl_2$, 5 mM β-mercaptoethanol and 10% ethanol). Subsequently, the bound proteins were eluted with step-wise increase of imidazole concentration (50, 100, and 200 mM) in a buffer consisting of 50 mM Tris–HCl, pH 8.0, 0.5M NaCl, 20 mM imidazole, 20% glycerol, 5 mM $CaCl_2$, 5 mM $MgCl_2$, 5 mM β-mercaptoethanol. Collected fractions were analyzed by SDS-PAGE with Coomassie blue staining and immunoblot analysis using anti-DGKα antibody (*Yamada, Sakane & Kanoh, 1989*).

Fractions containing full-length DGKα were further purified using size-exclusion chromatography on a Superdex 200 column 16/60 equilibrated with 20 mM Tris–HCl, pH 7.4 with 200 mM NaCl, 3 mM $CaCl_2$, 3 mM $MgCl_2$, 0.5 mM dithiothreitol, and 5% glycerol. Resulting fractions were analyzed by SDS-PAGE followed by Coomassie blue staining. Protein quantification was done by Bradford assay or using the extinction coefficient, $E_{0.1\%} = 1.14$. Gel filtration standards (Bio-Rad, Hercules, CA, USA) containing thyroglobulin (670 kDa), $\gamma$-globulin (158 kDa), ovalbumin (44 kDa), myoglobin (17 kDa), and vitamin B12 (1.3 kDa) were used to determine the molecular mass of proteins.

### In vitro DGKα activity assay

Activity of full-length DGKα was determined using the octyl-β-D-glucoside mixed micelle assay combined with the ADP-Glo™ kinase assay kit (Promega, Madison, WI, USA), as previously described (*Sato et al., 2013*; *Liu et al., 2016*). Briefly, the substrate micelle mixture containing 50 mM *n*-octyl-β-D-glucoside (Dojindo, Kumamoto, Japan), 10 mM (27 mol%) phosphatidylserine (PS; Sigma-Aldrich, St. Louis, MO, USA), 2 mM (5.4 mol%) 1,2-dioleolyl-*sn*-glycerol (DG; Sigma-Aldrich, St. Louis, MO, USA), 0.2 mM adenosine 5'-triphosphate (ATP) in a final buffer consisting of 50 mM MOPS, pH 7.4, 100 mM NaCl, 20 mM NaF, 10 mM $MgCl_2$, 1 µM $CaCl_2$ was mixed with 5 µL of purified DGKα to initiate enzymatic reaction. The reaction mixtures were incubated at 30 °C for 30 min. Subsequently, 25 µL of ADP-Glo reagent was added and incubated at room temperature for 40 min to terminate the enzyme reaction and deplete the remaining ATP. Kinase Detection Reagent (50 µL) was then added to convert the ADP produced from the kinase reaction into ATP for a luciferase/luciferin reaction. The reaction was performed at room temperature for 40 min and the luminescence from the luciferase/luciferin reaction was measured with a GloMax microplate reader (GloMax; Promega). A standard curve for ADP was generated by fitting a various concentration of ADP ranging from 25 to 200 µM and the corresponding luminescence signals relative luminescence unit by linear regression, and was used to convert the luminescence

intensities from DGKα reaction into ADP concentrations. To determine kinetic constants, the activity assay was performed under a series of concentrations of ATP (20 μM–1 mM) and DG (0–5.4 mol%), respectively. DGKα purified by size exclusion chromatography was added to 100 ng for each reaction and the assays were done in triplicate for each ATP or DG concentrations. The $K_m$ value was obtained by fitting the kinase activity of DGKα with the Michaelis–Menten equation using Prism 7 (GraphPad Software, La Jolla, CA, USA). To test the calcium dependency of DGKα activity, the enzyme activity was measured under the conditions containing either EGTA (3 mM) or $CaCl_2$ (0.6 mM).

### Inhibitor activity assay

Inhibitory activity of a previously identified inhibitor, CU-3 (*Liu et al., 2016*), against DGKα was measured with the octyl-β-D-glucoside mixed micelle assay followed by the ADP-Glo assay. A concentration series of CU-3 (0.02–10 μM) was incubated with the purified DGKα for 30 min at room temperature before adding to a reaction mixture for the assay. Half maximal inhibitory concentration (IC50) was determined by fitting the CU-3 dependent decrease of DGKα activity with the variable slope model in Graphpad Prism 7 software.

### Circular dichroism spectroscopy

Circular dichroism spectrum were recorded at ambient conditions between 190 and 250 nm on a Jasco J-805 spectrometer (JASCO Corporation, Tokyo, Japan) using a cell with path length of 0.2 mm, 20 nm/min scan speed and a bandwidth of 1 nm. DGKα was prepared at 0.32 mg/mL (3.75 μM) in 20 mM Tris–HCl buffer, pH 7.5, 10 spectra were averaged and a spectrum obtained for the buffer was subtracted. Spectral data were analyzed using the program Contin-LL (*Provencher & Glöckner, 1981*) suited in the DICHROWEB platform (*Whitmore & Wallace, 2004*).

## RESULTS

### A full-length form of DGKα was expressed in baculovirus-infected insect cells and purified as a monomer

We have previously reported that DGKα-CD possess enzymatic activity comparable to that of the full-length enzyme when expressed in COS-7 cells (*Sakane et al., 1996*), indicating that its substrate (ATP and DG) binding sites locate in the CD and DGKα-CD is an essential target for inhibitor development. Full-length DGKα also contains cysteine-rich C1 domains (Fig. 1A), which might be detrimental for correct folding in heterologous expression hosts. Therefore, we have first attempted to express DGKα-CD in *E. coli* by revamping the previous approach by *Petro & Raben (2013)*. In addition to N-terminal GST and MBP-tags, which were previously utilized (*Petro & Raben, 2013*), we have used Sumo domain fusion-tag for the enhancement of expression and solubility (*Butt et al., 2005*; *Marblestone et al., 2006*). To further increase the chance for expression of soluble proteins, we have also applied a multiple-construct approach (*Gräslund et al., 2008*) to prepare DGKα-CD constructs which have different N- and C-terminal boundaries (S332–G722, D344–G722, D369–G722, S332–S735, D344–S735, D369–S735). Each of those constructs was fused with the GST, MBP, and Sumo-tags. Despite our efforts, those constructs

resulted in either insoluble inclusion body formation (with GST-tag), or insufficient translation and proteolytic degradation (with MBP-tag), or soluble microscopic aggregation (with Sumo-tag) (Fig. S1).

To circumvent the difficulty associated with bacterial expression system, we have used baculovirus-infected Sf9 cells to produce DGKα-CD. The construct of DGKα-CD (D364–S735) with N-terminal His$_{\times 6}$ tag was cloned into pOET3 transfer vector harboring the late AcMNPV p6.9 promoter, which provides earlier expression compared to the polyhedrin promotor. The recombinant DGKα-CD was expressed in cultured insect cells using the flashBAC baculovirus vector expression system, and subsequently purified from cell-lysates using Ni-affinity chromatography (Fig. S2A). Following size-exclusion chromatography on a Superdex 200, however, demonstrated that DGKα-CD formed soluble aggregates eluting in the void volume of the column (Fig. S2B).

In our early studies, a native form of full-length DGKα has been purified from porcine thymus and this full-length form was found to be catalytically competent (*Sakane, Yamada & Kanoh, 1989*; *Sakane et al., 1991*). We therefore set to produce full-length DGKα (aa 1–735) using the same baculovirus expression system used for DGKα-CD. As expected, the vast majority of DGKα remained in soluble form after cell lysis, as shown by immunoblot analysis (Fig. 1B). Ni-affinity chromatography was conducted to purify DGKα from the cell lysis supernatant, and relatively pure DGKα was eluted in fractions containing 50 and 100 mM imidazole (Fig. 1C). To further purify DGKα, we next performed size-exclusion chromatography on a Superdex 200 column. Because DGKα bears calcium-binding EF-hand motifs and a magnesium ion was predicted to bind to the CD (*Abe et al., 2003*), we added 3 mM CaCl$_2$ and 3 mM MgCl$_2$ in the equilibration buffer. DGKα eluted as a single peak at 73.5 mL retention volume (Fig. 1D), which corresponds to the molecular mass of 77 kDa, based on a calibration curve obtained with molecular mass standard proteins. This result indicates that DGKα exists as a monomer in solution. DGKα was purified to near homogeneity (Fig. 1E) and the yield was approximately 1.3 mg per one L of Sf9 cell culture.

## Kinase activity assay and inhibitory assay for the purified DGKα

To test whether the purified DGKα is catalytically active, we conducted the octyl-β-ᴅ-glucoside mixed micelle assay combined with a luminescence-based assay that measures ADP produced in a kinase reaction (*Sato et al., 2013*; *Liu et al., 2016*). DGKα purified from the size-exclusion chromatography was found to exhibit kinase activity with peak fractions having the maximum activity (Fig. 2A). We have previously demonstrated that DGKα activity, which has been purified from porcine thymus or expressed in COS-7 cells, is enhanced by Ca$^{2+}$ binding to its two N-terminal EF-hand motifs (*Sakane et al., 1990*, *1991*; *Yamada et al., 1997*). As predicted, the purified DGKα exhibited significantly reduced activity when the bound calcium ions were chelated with 3 mM EGTA (Fig. 2B). Furthermore, no significant changes of the activity were observed after storage of the purified DGKα at 4 °C for at least 3 months.

We also determined the kinetic parameters of DGKα for ATP and DG to assess the catalytic properties of the purified DGKα. ATP-dependent increase of the kinase activity

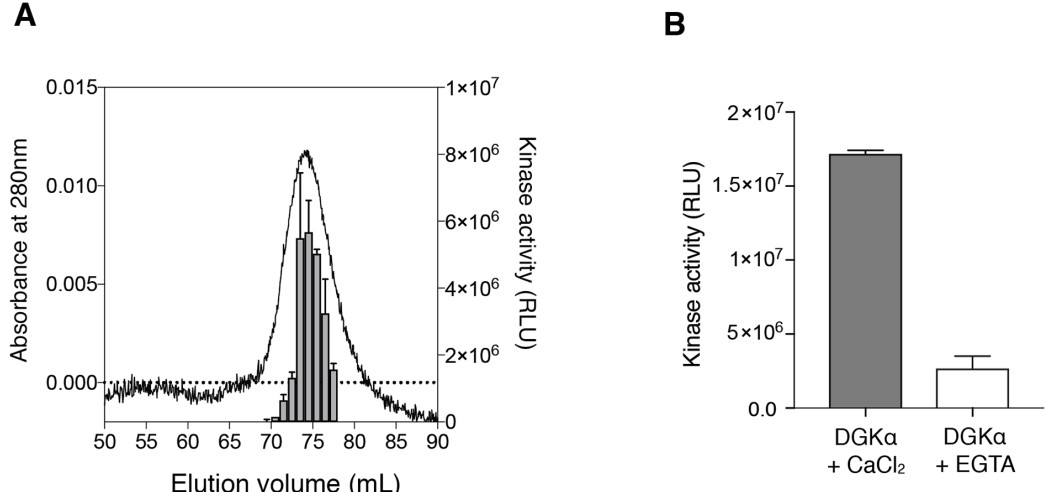

**Figure 2 Purified DGKα is catalytically active and positively regulated by Ca²⁺.** (A) Luminescence-based (ADP-glo) kinase activity assay of fractions from size exclusion chromatography of DGKα. Five microliters from each fraction containing 38.5–363 ng of DGKα was added for a reaction and the following details are described in "Materials and Methods." Luminescence values are presented as relative luminescence unit (RLU) over background signals from a well containing a buffer (20 mM Tris–HCl, pH 7.4, 0.2M NaCl, 3 mM CaCl$_2$, 3 mM MgCl$_2$, 0.5 mM DTT, and 5% glycerol) used for size-exclusion chromatography. (B) Calcium-dependent activity of the purified DGKα. The luminescence-based DGK activity assay was conducted using 150 ng of DGKα in the presence of CaCl$_2$ (0.6 mM) and EGTA (3.6 mM). Purified DGKα was pre-incubated with 3 mM EGTA for 30 min on ice to chelate CaCl$_2$ contained in a buffer used for size exclusion chromatography, and concentrated EGTA was also added into the reaction mixture at a final concentration of 3.6 mM. Measured luminescence values of DGKα in the presence of CaCl$_2$ or EGTA were subtracted with each negative control (CaCl$_2$ or EGTA) and data shown are mean ± SD for triplicate measurements.

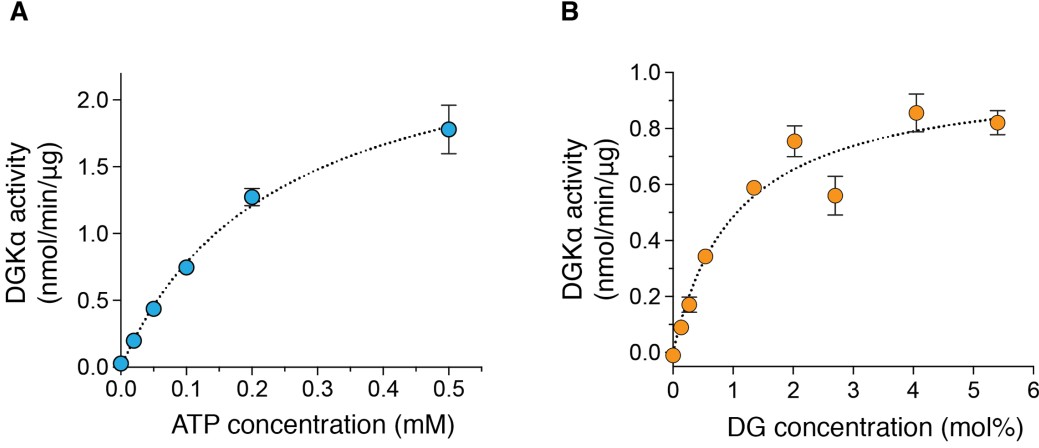

**Figure 3 The enzymatic kinetics of the purified DGKα with ATP and diacylglycerol (DG).** (A) ATP dependency of DGKα activity was measured with the luminescence based assay. (B) DGKα activity was plotted as a function of DG concentration (mol%). Measured luminescence values were converted into the amount of ADP produced (nmol) based on the ATP-to-ADP conversion curve separately measured with known concentration of ATP (50 μM–1 mM). Data shown are mean ± SD for triplicate measurements.

**Table 1 Enzyme kinetic parameters of DGKα with ATP and diaclyglycerol.**

| Substrate | $K_m$ | $V_{max}$ |
| --- | --- | --- |
| ATP | $0.24 \pm 0.03$ mM | $2.66 \pm 0.15$ nmol/min/μg |
| Diacylglycerol | $1.06 \pm 0.21$ mol% | $1.00 \pm 0.06$ nmol/min/μg |

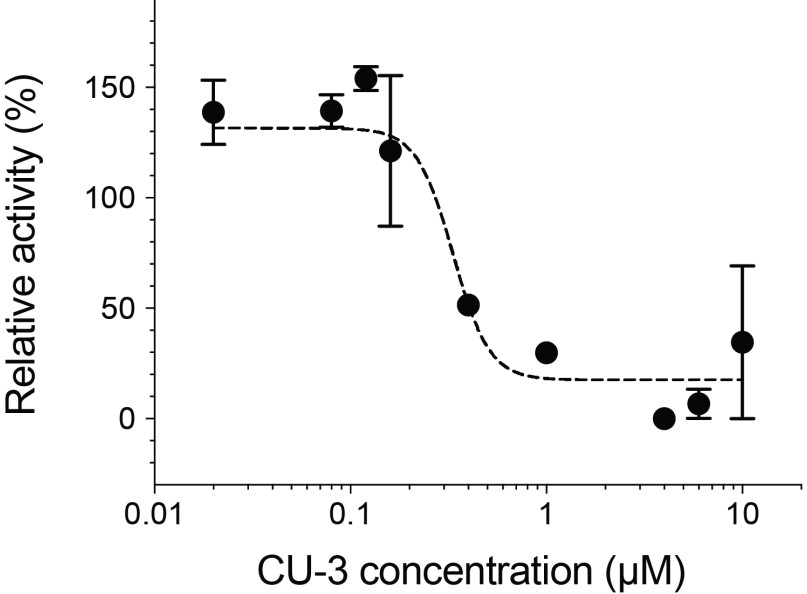

**Figure 4 Inhibitory activity of a small molecule inhibitor, CU-3.** A concentration series of CU-3 ranging from 0.02 to 10 μM was incubated with the purified DGKα (100 ng) for 30 min at room temperature before adding to a reaction mixture for the mixed micelle/luminescence-based assay. Half maximal inhibitory concentration ($IC_{50}$) was determined by fitting the CU-3 dependent decrease of DGKα activity with the variable slope model. In the absence of DGKα, luminescence signals with various concentrations of CU-3 were negligible and no dose-dependent changes were observed.

was observed (Fig. 3A) and the $K_m$ value was determined to be $0.24 \pm 0.03$ mM (Table 1), comparable with those obtained with DGKα from porcine thymus (0.1 mM) (*Sakane et al., 1991*) or DGKα expressed in COS-7 cells (0.1–0.25 mM) (*Sato et al., 2013*; *Liu et al., 2016*). The activity was also increased in a DG-concentration dependent manner (Fig. 3B) and the $K_m$ value of $1.1 \pm 0.21$ mol% (Table 1) was consistent with those from our previous studies (3.3 mol% with DGKα purified from porcine thymus (*Sakane et al., 1991*), 1.9–3.4 mol% with DGKα expressed in COS-7 cells (*Sato et al., 2013*; *Liu et al., 2016*)). For both cases, compared to our previous study using crude lysates of mammalian cells (*Sato et al., 2013*; *Liu et al., 2016*), the relative activity increased nearly 50-fold when the purified DGKα was used. Furthermore, the kinase activities of our purified DGKα (1 to 2 nmol/min/μg) is comparable to those obtained with DGKα from porcine thymus (2.4 nmol/min/μg) (*Sakane et al., 1991*). These results demonstrate that the purified DGKα is in a fully functional state and stable during purification and storage.

We next measured the inhibitory activity of CU-3, a previously identified DGKα inhibitor (*Liu et al., 2016*). CU-3 is an ATP competitive inhibitor with an $IC_{50}$ value

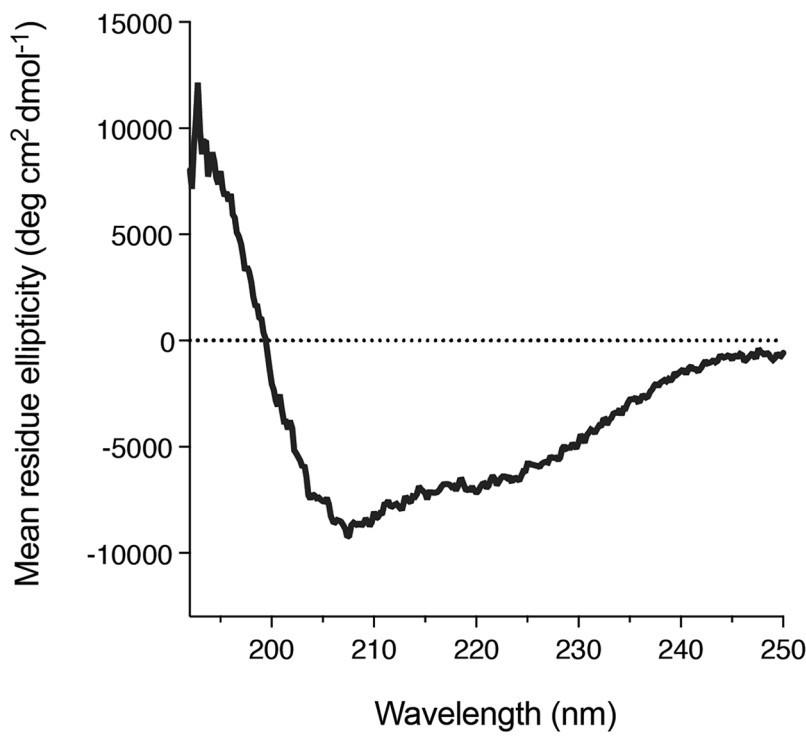

**Figure 5 Secondary structure of the purified DGKα.** Circular dichroism spectrum of DGKα measured at ambient conditions between 190 and 250 nm on a Jasco J-805 spectrometer. DGKα was prepared at 0.3 mg/mL in 20 mM Tris–HCl buffer, pH 7.4, 150 mM NaCl, 3 mM MgCl$_2$, 3 mM CaCl$_2$, 5 % glycerol. The analysis of the circular dichroism spectrum using the program Contin-LL (*Provencher & Glöckner, 1981*) suited in the DICHROWEB platform (*Whitmore & Wallace, 2004*) showed the presence of both α-helical (18.9 %) and β-strand (27.4 %) structures.

of 0.6 μM. Consistent with this, CU-3 inhibited the activity of DGKα in a concentration-dependent manner with an IC$_{50}$ value of 0.34 ± 0.1 μM (Fig. 4).

## Structural characterization of the purified DGKα

We also found that DGKα solution could be concentrated using a centrifugal filter without any significant loss of the protein. Concentrated DGKα remained as a monomer as demonstrated by a size-exclusion chromatography (Fig. S3). Using the concentrated DGKα (0.32 mg/mL), we characterized the secondary structure using circular dichroism spectroscopy. The circular dichroism spectrum of DGKα and following analysis indicates that DGKα is well-folded and contains certain amounts of α-helical (18.9%) and β-strand (27.4%) structures (Fig. 5), further demonstrating that the expression of a full-length DGKα, not a solo CD, in the baculovirus-infected insect cells is suitable for producing a natively folded and active form of DGKα.

## DISCUSSION

Diacylglycerol kinases are a family of multi-domain lipid kinase that regulate a variety of cellular process (*Sakane et al., 2007*; *Merida, Ávila-Flores & Merino, 2008*; *Shulga, Topham & Epand, 2011*), and DGKα has recently emerged as a novel therapeutic

target for cancer immunotherapy (*Dominguez et al., 2013*; *Purow, 2015*; *Sakane, Mizuno & Komenoi, 2016*; *Liu et al., 2016*; *Noessner, 2017*). However, no structural information of DGKs, especially their CD, is available. This is largely because the procedure for large scale production of recombinant DGKs in their soluble and homogeneous form, a prerequisite for protein crystallization, is not well-established. Here we have used the baculovirus-insect cell expression system to produce a full-length form (DGKα), and investigated the enzymatic and structural properties in vitro.

*Petro & Raben (2013)* have made significant efforts to express and purify a pig DGKα and DGKα-CD using bacterial expression system with several fusion tags (GST, TRX, and MBP), a set of bacterial chaperons, or in vitro refolding. Despite their pursuit, expressed DGKα constructs either formed inclusion bodies or soluble microscopic aggregates. We have also used *E. coli* cells to produce DGKα-CD with several N-terminal fusion tags (GST, Sumo, and MBP) and with different N- and C-terminal boundaries. While both MBP- and Sumo-fused DGKα-CD remained in a soluble fraction after cell-lysis and Ni-affinity chromatography (Figs. S1A and S1B), those DGKα-CD with fusion-tags eluted in the void volume of Superdex 200 column (Fig. S1C). When expressed with MBP, an elution fraction from the Ni-affinity chromatography also contained additional smaller bands along with MBP-fused DGKα-CD (Fig. S1A), which could be due to insufficient translational ability of *E. coli* for producing eukaryotic proteins, as previously suggested (*Petro & Raben, 2013*). Because baculovirus-insect cell expression system has both the capacity to produce recombinant proteins at a large scale and the capability to provide eukaryotic protein expression machineries, we next utilized this system to produce DGKα-CD. The protein expressed in Sf9 cells was soluble, however, contrary to our expectation, the protein formed soluble aggregates, which eluted in the void volume (Fig. S2). These results indicate that the only CD has a tendency to self-aggregate upon isolation, possibly due to its intrinsic characteristics that recognize DG embedded in plasma membrane, and suggest that the only CD is not suitable for structural studies even if it is expressed using a eukaryotic expression system.

In contrast to the CD, full-length DGKα elutes in a relatively sharp peak of size-exclusion chromatography and remains as a monomer when it is assumed to have a globular shape (Fig. 1C; Fig. S3). Such production of a full-length DGKα in a soluble and monomeric form using the baculovirus insect cell expression system holds promise for the preparation of DGKα sample suitable for protein crystallization screening. DGKα consists of the N-terminal regulatory domains including EF-hand motifs and C1 domains, and the C-terminal CD. This suggests that DGKα adopts a compact globular structure rather than an elongated one. YegS (a putative bacterial lipid kinase) (*Bakali, Nordlund & Hallberg, 2006*), a bacterial DgkB (*Miller et al., 2008*), and a human sphingosine kinase (SphK1) (*Wang et al., 2013*) have been successfully purified and their crystal structures have been reported (*Bakali et al., 2007*; *Miller et al., 2008*; *Wang et al., 2013*). Although all of those lipid kinases are homologous to mammalian DGKs and belong to a protein family PF00781 (*DAGK_cat*), they do not possess N-terminal

regulatory domains. This might explain why the N-terminal domain of DGKα is required to obtain the protein as a soluble monomer. Interestingly, previous studies by us and others have suggested the presence of intramolecular interactions between the N-terminal regulatory domains and the CD (*Merino et al., 2007*; *Takahashi et al., 2012*). It is reasonable to surmise that a potential aggregation-prone surface of the CD of DGKα is intra-molecularly masked by the N-terminal regulatory domains including recoverin homology, EF-hand motif, and C1 domains.

Enzymatic characterization of DGKα reveals that $K_m$ values to ATP (0.24 mM) and DG (1.1 mol%) are very similar to those obtained using DGKα partially purified from porcine thymus (0.1 mM for ATP and 3.3 mol% for DG, respectively) (*Sakane et al., 1991*) or DGKα expressed in COS-7 cells (0.1–0.25 mM for ATP and 1.9–3.4 mol% for DG, respectively) (*Sato et al., 2013*; *Liu et al., 2016*), further demonstrating the effectiveness of baculovirus insect cell expression system for producing DGKα not only in soluble and homogeneous form, but also in its active one.

In summary, this study demonstrates that the production of full-length DGKα, not DGKα-CD alone, using the baculovirus-insect cell expression is a very promising approach to produce DGKα samples for future in vitro structural and functional studies. Firstly, DGKα has been purified by Ni-affinity and size-exclusion chromatographies to near-homogeneity, and purified DGKα remains in soluble and monomeric form, and can be concentrated without any significant loss of the protein, which are prerequisites for protein crystallization. Purified DGKα sample, however, still contains slight amounts of contaminant proteins which might non-specifically bind to DGKα. Further modification and optimization of the protein construct and purification conditions must be required. Secondly, the obtained yield of DGKα, 1.3 mg per one L cell culture, is enough to initiate crystal screening. Thirdly, the purified DGKα is catalytically competent. The measured kinase activity and the $K_m$ values to ATP and DG are comparable to those obtained with native form of DGKα partially purified from porcine thymus and DGKα expressed in mammalian cells.

## CONCLUSION

We demonstrate that the baculovirus-insect cell expression of the full-length form of DGKα, not DGKα-CD alone, represents a promising approach to produce protein sample suitable for structural studies of DGKα. We believe that this study will encourage future pursuits to determine crystal structures of mammalian DGKs that has still remained enigmatic for almost 60 years since its identification (*Hokin & Hokin, 1959*).

## ACKNOWLEDGEMENTS

We thank Dr. Saurav Misra from Kansas State University for helpful suggestions regarding this work. We thank Dr. Haobo Jiang from Oklahoma State University for providing pSUMO vector. We thank Dr. Naoto Yonezawa (Chiba University) for giving Sf9 cells. We are grateful to Dr. Takeshi Murata (Chiba University) for the use of their BioLogic chromatography system.

### Funding

This work was supported in part by JSPS KAKENHI (Grant Numbers: 17K115444 to Daisuke Takahashi and 26291017, 15K14470, 17H03650 to Fumio Sakane), Association of Graduate Schools of Science and Technology in Chiba University (Daisuke Takahashi), and the Futaba Electronic Memorial Foundation; the Ono Medical Research Foundation; the Japan Foundation for Applied Enzymology; the Food Science Institute Foundation; the Skylark Food Science Institute; the Asahi Group Foundation and the Japan Milk Academic Alliance (Fumio Sakane). The funders had no role in study design, data collection and analysis, decision to publish, or preparation of the manuscript.

### Grant Disclosures

The following grant information was disclosed by the authors:
JSPS KAKENHI: 17K115444; Association of Graduate Schools of Science and Technology in Chiba University (Daisuke Takahashi).
JSPS KAKENHI: 26291017, 15K14470, 17H03650; Futaba Electronic Memorial Foundation; the Ono Medical Research Foundation; the Japan Foundation for Applied Enzymology; the Food Science Institute Foundation; the Skylark Food Science Institute; the Asahi Group Foundation and the Japan Milk Academic Alliance (Fumio Sakane).

### Competing Interests

The authors declare that they have no competing interests.

### Author Contributions

- Daisuke Takahashi conceived and designed the experiments, performed the experiments, analyzed the data, contributed reagents/materials/analysis tools, prepared figures and/or tables, authored or reviewed drafts of the paper, approved the final draft.
- Fumio Sakane conceived and designed the experiments, analyzed the data, contributed reagents/materials/analysis tools, authored or reviewed drafts of the paper, approved the final draft.

### Data Availability

The raw data are provided in the Supplemental Files.

### Supplemental Information

Supplemental information for this article can be found online at http://dx.doi.org/10.7717/peerj.5449#supplemental-information.

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
