# Peer review of "Expression and purification of human diacylglycerol kinase α from baculovirus-infected insect cells for structural studies"

_PeerJ, doi:10.7717/peerj.5449_

## Round 0.1 · original submission · Minor Revisions

This work scientifically sounds, it is well written and it has been carried out with great care. There are minor points highlighted by the referees. Adding the requested experiments would definitively be beneficial

Reviewer 1 ·

Basic reporting

Takahashi and Sakane have used a mix of experiments to assess the production of diacylglycerol kinaseα (DGKα), which is considered a potential target for drug discovery in cancer. Their investigation is especially focused on the purification of the enzyme from baculovirus-infected insect cells. This study might represent a promising approach to obtain the three-dimensional structure of the protein. The authors conclude that the protein is enzymatically active, soluble and homogeneous.

The manuscript is clearly written with the introduction and the references relevant to the study. However, I would like to suggest some editing of the manuscript, where I found some mistypos. For example, in several sentences the word "was" should be changed in "where" as plural is used. Furthermore, the language could be slightly improved in lines 94, 157, 316-317.

The figures are relevant to the work presented and very well labelled and described. However, Figures 1C and E, S1a and b, S2A could be improved to better show the homogeneity in the purification steps (i.e. through a TCA/acetone precipitation or Silver staining before performing the SDS-page analysis).

I thank you for providing the raw data, however I noticed that the data relative to Fig 4. (Inhibitory activity of CU-3) represent a single experiment and reported in Relative activity (%) and not as raw data. It would be useful to carry out an assay in duplicates or triplicates. Furthermore, the future readers would benefit in having the raw data from the kinetic assay and the transformed ones for comparison.

Experimental design

The methods are well described and meaningful.
However, with regard to the kinetic and inhibitory assays, adding the negative controls (with no DGK and no DGK in presence of EGTA) to the figures 2B, which are missing, will strongly support the statement about the enzymatic activity of DGK. The kinetic assay with CU-3 should be carrying out also with the negative controls (with no DGK) to exclude any side reaction with the ADP-glo kit (most important issue).

Validity of the findings

1. The authors assert that they obtained the protein to near-homogeneity and in a soluble form, and it can be concentrated without any loss of the protein.
To confirm this statement, the loaded fractions in the Figures 1C and 1E should be concentrated, using the Silver staining or the TCA/acetone or Methanol/Chloroform precipitation approach, as the protein bands appear too thin. Furthermore, as the full-length DGK contains 12 Tryptophans, 21 Tyrosines and 25 Phenylalanines, it is not clear why in the figure 1D, 2A and S3, the absorbance at 280 nm is too low. As the authors claim that the protein could be concentrated and it is not aggregating during this step, they should perform the gel filtration chromatographies with concentrated samples, which will clarify the homogeneity and help the future readers for the crystallization purpose.

2. The authors suggest that DGK with a molecular weight of ~ 85kDa is around 77 kDa when performing gel filtration analysis. In order to have better resolution and consistent data, I would advise to perform an analytical gel filtration instead.

3. Figure 2B is very nice, however I would suggest adding the negative controls, as ADP-glo assay is a coupled assay.

4. For the kinetic assay and to easy read the results, I would propose adding a table with the Km (in mM for both ATP and DG), Kcat and Vmax, with the relative standard errors, which are missing in the manuscript.

5. For the inhibitory activity of CU-3, the experiment should be implemented at least in duplicate, as it is reported as single experiment.
IC50 value should be reported with its standard error.
Most importantly, the negative control is missing. The assay in presence of CU-3 should be carrying out without DGK, to exclude that CU-3 is not a PAINs compound (Baell J and Walters MA, Nature 2014). The compound has phenol-sulphamide and iso-thiazolone moieties, which might be both redox cycler and covalent modifier. For the reactions involved in ADP-glo assay kit, it is kindly recommended to check the inhibitory activity of this compound against DGK.

Additional comments

I commend the authors for their extensive work in finding the conditions for the production and purification of a challenging protein.
Soluble proteins have also been obtained from bacterial expression. As an extensive work was done, the expression could be optimised with an IPTG titration or auto induction approach.

Conclusions are well stated and linked to the original research question.

·

Basic reporting

No comment

Experimental design

No comment

Validity of the findings

No comment

Additional comments

The authors provide for the first time an experimental design that allows purification of the full length DGKalpha isoform. Kinetics studies, calcium dependance and inhibitor sensitivity correlate with those reported previously by the group and other laboratories. The article demonstrates the expertise of the teal on this particular area where have contributed with important findings. in this ocasiion, the purification of recombinant DGKalpha opens the possibility for long awaited structual studies

·

Basic reporting

no comment

Experimental design

no comment

Validity of the findings

no comment

Additional comments

Comments to the author:

This is an interesting investigation done by Sakane group. Author has done extensive work to optimize the expression and purification of an important human diacylglycerol kinase α enzyme. Several constructs have been made and expression was tested in E. coli and insect cell line expression system. Author makes the point that catalytic domain of the DGKα form soluble oligomer whereas full length enzyme is stable and catalytically active. In vitro assays of the full length DGKα shows that purified enzyme is active and can be used for structural characterizations. The study has been carried out with great care and caution, however, some issues demand proper attention by the authors.

1. Line 285 and 288: In the enzyme activity assay, it seems author has reported the Km for the ATP and DG but the standard error is not reported.

2. Line298: The IC50 is reported without standard error.

3. Inhibition assay: In the raw data, the DGKα with zero concentration of CU-3 has relative activity of 100% as control. But 0.02, 0.04, 0.08 and 0.16 uM CU-3 treated enzyme has relative activity higher than 100%. I am wondering, what is explanation for this higher activity? If possible, I would encourage the author to repeat the experiment as points on the plot (Fig4) are a bit scattered.

4. No figure legends are provided for supplementary figures.

5. Line 273: correction; To test whether the purified DGKα is biologically active, ……. Should be “To test whether the purified DGKα is catalytically active,………”


Clear and unambiguous; and professional English used throughout in the manuscript. I would recommend this important piece of work for the publication in PeerJ after the raised issues are taken care.

---

## Round 0.2 · accepted · Accept

The authors amended the manuscript and improved the data obtained; the manuscript is worthy of publication.

# Reviewer 1 ·

Basic reporting

Takahashi and Sakane performed several experiments for the production of diacylglycerol kinaseα (DGKα), a potential target for drug discovery in cancer. Their investigation, especially focused on the purification of the enzyme from baculovirus-infected insect cells, is a promising approach for future structural studies. The authors conclude that the protein is enzymatically active, soluble and homogeneous.

The manuscript is clearly written with the introduction and the references relevant to the study. Corrections and new raw data would definitively be beneficial for other scientists.
The addition of standard errors for the activity and inhibition assays is very helpful, as well the table which would be beneficial for other readers.
I commend the authors for their extensive work in finding the conditions for the production and purification of this protein and for the effort made in revising the manuscript.

Conclusions are well stated and linked to the original research question.

Experimental design

The methods are well described and meaningful.

The addition of negative controls to the kinetic and inhibitory assays makes the investigation rigorous and makes the research question well defined.

Validity of the findings

1. The authors assert that protein samples from affinity chromatography and size-exclusion chromatography show homogeneity in SDS-page and they decide to retain the Figure 1C and 1E ( and the supplementary one). However, in the gel the bands relative to the enzyme are too thin and some contaminants could be clearly seen below the DDGKα band.
Protein precipitation (i.e TCA/acetone, Ethanol or Methanol/chloroform) protocols are not only used to remove detergents, salts or lipids: they are useful to concentrate a diluted sample. Furthermore, silver staining will definitively show contaminants. For crystallisation purpose, the protein should be seen as strong band in a SDS-page (which I do not see), and often two chromatography steps are not enough to obtain a pure sample.
Furthermore, as the authors claim the protein could be concentrated, they could have performed a SDS-page with a concentrated protein without using TCA/acetone precipitation approach. This would have clarified the homogeneity and helped the future readers for the crystallisation.

2. The authors affirm their size-exclusion chromatography is clear enough to show the monomeric form of DGKα. The column superdex 200 16/60 is not always able to discriminate between different forms (i.e. dimer/monomer). The reason why I have suggested the use of analytical gel filtration superdex 200 10/300 gl increase relies on the fact that this new generation column allow higher resolution and high-quality data. I do not exclude that their results with GF 16/60 suggest a monomeric state of the protein.

Additional comments

I commend the authors for their extensive work in finding the conditions for the production and purification of this enzyme.
Surely, the expression of this protein could be optimised from the bacterial system described in the manuscript.

·

Basic reporting

no comment

Experimental design

no comment

Validity of the findings

no comment

Additional comments

I am happy with the revision authors have made. It is an excellent paper and worthy of publication in PeerJ.